# Pain perception genes, asthma, and oral health: A reverse genetics study

**Rosany O. Lisboa**[1,2,3ᵒ], **Raymond F. Sekula**[4‡], **Mariana Bezamat**[2ᵒ], **Kathleen Deeley**[2‡], **Luiz Carlos Santana-da-Silva**[1,3‡], **Alexandre R. Vieira**[2ᵒ]*

**1** Laboratory of Inborn Errors of Metabolism, Institute of Biological Sciences, Federal University of Pará, Pará, Brazil, **2** Departments of Oral and Craniofacial Sciences, Pediatric Dentistry and Center for Craniofacial and Dental Genetics, School of Dental Medicine, University of Pittsburgh, Pittsburgh, Pennsylvania, United States of America, **3** Graduate Program in Oncology and Medical Sciences, Federal University of Pará, Pará, Brazil, **4** Department of Neurological Surgery, Columbia University Vagelos School of Medicine, New York, New York, United States of America

ᵒ These authors contributed equally to this work.
‡ These authors also contributed equally to this work
* arv11@pitt.edu

**Data Availability Statement:** All relevant data are within the manuscript and its Supporting information files.

## Abstract

Pain is an experience of a subjective nature, interpreted in a personal way and according to an extensive palette of factors unique to each individual. Orofacial pain can be acute or chronic and it is usually the main reason for the patient to seek dental care. Pain perception varies widely among individuals. This variability is considered a mosaic of factors, which include biopsychosocial factors and genetic factors. Understanding these differences can be extremely beneficial for pain management in a personalized and more efficient way. We performed association studies to investigate phenotypes associated with genetic markers in pain-related genes in two groups of patients who received more or less anesthesia during dental treatment. The study group was comprised of 1289 individuals participating in the Dental Registry and DNA Repository Project (DRDR) of the University of Pittsburgh, with 900 participants in the group that received the most anesthesia and 389 constituting the comparison group that received less anesthesia. We tested 58 phenotypes and genotypic data of seven SNPs in genes that are associated with pain perception, pain modulation and response to drugs used in pain treatment: *COMT* (rs4818 and rs6269), *GCH1* (rs3783641), *DRD2* (rs6276), *OPRM1* (rs1799971), *SCN9A* (rs6746030) and *SCN10A* (rs6795970). The analysis revealed a protective effect of rs1799971 on asthma in the total sample. rs3783641 was associated with salivary secretion disorders in females who received more anesthesia. rs1799971 was also associated with periodontitis in Whites who received less anesthesia. rs4818 was associated with disease and other tongue conditions in the group composed of Blacks who received less anesthesia. In conclusion, our study implicated variants in pain-related genes in asthma and oral phenotypes.

**Funding:** The authors received no specific funding for this work.

**Competing interests:** The authors have declared that no competing interests exist.

## Introduction

Pain in the region above the neck, in front of the ears and below the orbitomeatal line, including the oral cavity, are considered orofacial pains [1]. Due to the complexity of these regions, orofacial pain can arise from several sources and represents a challenge for diagnosis and clinical treatment [2].

Nociceptive pain, which is characterized by the acute stimulation of nociceptors found in the skin, intraoral cavity, and dental pulp, is considered the most common type of pain in the orofacial region and the reason why patients usually seek dental care [3].

Despite the discrepancies between studies, most point out that sensitivity to pain is more prevalent in females than males. Possible explanations include the effects of gonadal hormones, genetic factors, and differences in the activation of brain patterns [4]. Some studies also suggest that descending pain inhibitors, such as those that induce opioid and non-opioid analgesia, also appear to be influenced by sex, and gonadal hormones [5,6].

Studies on orofacial pain related to age are usually carried out with patients suffering from chronic orofacial pain. Boggero et al. (2015) [7] demonstrated that disruption of individual's daily activities caused by the presence of high levels of pain intensity decreased with age, supporting the idea that older adults deal with pain as well or better than younger adults.

The experience of pain is multifactorial, with the influence of genetic factors, sex, socioeconomic status and access to an adequate health system [8,9].

Pain cannot be strongly associated with the degree of damage or inflammation observable by clinicians [3]. Therefore, awareness that different patients might often report completely different degrees of pain perception is essential for a mechanistic understanding of orofacial pain. This knowledge will provide better identification of individuals who are at risk of not responding to conventional interventions to treat pain [4]. Thus, three main factors are involved in minimizing orofacial pain during common dental treatments: 1) increased knowledge in the efficiency of pharmacological agents, 2) improved injection techniques, and 3) the identification of differences inherent to each individual that may indicate the need for more or less anesthesia.

In order to detect associations of distinct oral and systemic diseases, we used individual indication of the need for more anesthesia to define comparators. Then, a "reverse genetics" approach was used and a genotype-phenotype strategy to assess the association between single nucleotide polymorphisms (SNPs) and a wide range of phenotypic variables in a high-throughput manner was implemented [10]. Our objective was to search for associations between clinical phenotypes and genetic markers in genes related to pain.

## Materials and methods

### Subjects

From 5,025 individuals, part of the University of Pittsburgh School of Dental Medicine Dental Registry and DNA Repository (DRDR) project, a total of 1,289 subjects were selected for this study. These individuals were selected because they received more than one anesthetic carpule (more than 1.8 cc of volume per tube) and constituted the first group (n = 900), or received a smaller amount of anesthesia that varied from half to one tube, forming the second group (n = 389). Those two groups were matched by sex, age, self-reported ethnicity, and dental treatment performed. The sample in terms of sex, ethnicity, and age is summarized in Table 1. The overall study design is illustrated in Fig 1.

The University of Pittsburgh Institutional Review Board (IRB # 0606091) approved this project. Written informed consent documents were obtained from all subjects. Age-

**Table 1. Description of the study sample.**

| Variables | More Local Anesthetic (n = 900) | | Less Local Anesthetic (n = 389) | |
|---|---|---|---|---|
| **Age in years** | | | | |
| (average ± standard deviation, range) | 45.5 ± 17.34 | (8–85) | 45.3 ± 18.24 | (10–87) |
| **Sex (n, %)** | | | | |
| Female | 513 | (57%) | 206 | (53%) |
| Male | 387 | (43%) | 183 | (47%) |
| **Self-reported Ethnicity** | | | | |
| White | 695 | (77%) | 297 | (77%) |
| Black | 161 | (18%) | 70 | (18%) |
| Asian | 27 | (3%) | 19 | (5%) |
| Others | 17 | (2%) | 3 | (1%) |

appropriate assent documents were used for children younger than 14 years of age and signed informed consent documents were obtained from the parents.

This study is characterized as a human observational study that complies with the Strengthening the Reporting of Observational Studies in Epidemiology (STROBE) guidelines. We attested that the STROBE checklist was completed.

## Phenotypes

This study includes both oral and general health related phenotypes, comprising inflammatory conditions that underlie orofacial pain processes. The internal DRDR diagnostic codes were converted into "phecodes" (high throughput phenotyping tool based on ICD—International Classification of Diseases—codes to make them compatible with the software used). Phenotypes tested are listed below with their respective phecode in parentheses: Phenotypes tested included candidiasis (112), cancer of mouth (145), hemangioma and lymphangioma any site (228), disorders of parathyroid gland (252), lymphadenitis (289.4), sleep related movement disorders (327.7), disorders of lacrimal system (375), varicose veins (454), chronic tonsillitis

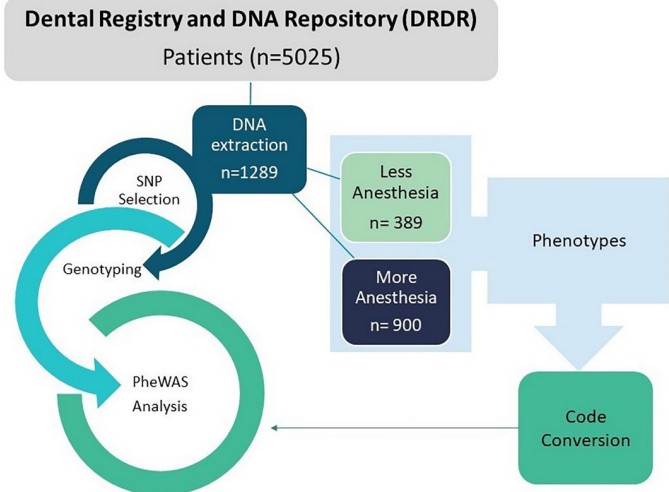

**Fig 1. Overall study design flowchart.** The stages of the study are presented from the selection of groups from the Dental Registry and DNA Repository (DRDR) to the PheWAS analysis.

and adenoiditis (474.2), disorders of tooth development (520), disturbances in tooth eruption (520.2), diseases of hard tissues of teeth (521), dental caries (521.1), diseases of pulp and periapical tissues (522), periapical abscess (522.5), gingival and periodontal diseases (523), gingivitis (523.1), periodontitis (acute or chronic) (523.3), acute periodontitis (523.31), chronic periodontitis (523.32), anomalies of tooth position/malocclusion (524.3), other diseases of the teeth and supporting structures (525), loss of teeth or edentulism (525.1), diseases of the jaws (526), cysts of the jaws (526.1), anomalies of jaw size/symmetry (526.3), temporomandibular joint disorders (526.4), exostosis of jaw (526.8), diseases of the salivary glands (527), disturbance of salivary secretion (527.7), diseases of the oral soft tissues, excluding lesions specific for gingiva and tongue (528), stomatitis and mucositis (528.1), stomatitis and mucositis (ulcerative) (528.11), diseases of lips (528.5), leukoplakia of oral mucosa (528.6), diseases and other conditions of the tongue (529), glossitis (529.1), other specified erythematous conditions (695.8), Diseases of sebaceous glands (706), Congenital anomalies of face and neck (749), epilepsy, recurrent seizures, convulsions (345), cerebral artery occlusion, with cerebral infarction (433.21), asthma (495), tuberculosis (10), other deficiency anemia (281), chronic sinusitis (475), chronic hepatitis (70.4), chronic liver disease and cirrhosis (571), spontaneous ecchymoses (287.1), cardiac dysrhythmias (427), rheumatic fever/chorea (41.21), hypertension (401), nonrheumatic mitral valve disorders (395.1), abnormal heart sounds (396), valvular heart disease/heart chambers (747.12), heart valve replaced (395.6), chronic renal failure [CKD] (585.3), and complication of internal orthopedic device (858).

A PheWAS package, installed into R Studio software was used (Carroll et al., 2014). Phecodes as well as genotypes for each patient were uploaded into the system. The software compares individuals for each phenotype according to their genotypes. An additive genetic model was implemented where allele frequencies were calculated and compared between each group.

## SNP (Single Nucleotide Polymorphism) selection and genotyping

SNPs were selected from previous studies reporting genes associated with pain perception and/or biological roles related to pain response, or analgesia. Table 2 contains a list of selected genes and SNPs as well as their minor allele frequencies (MAF) and their relationship with pain as previously demonstrated.

Participants' DNA were extracted and genotyped using established protocols previously described [21]. Genotyping of SNPs was performed using end-point analysis and TaqMan chemistry in 3.0 µl volumes on an ABI QuantStudio™ 6 Flex Real Time PCR System with a software version 1.7 (Applied Biosystems, Foster City, CA, USA).

## Statistical methods and power analysis

The standard statistical test in the PheWAS package is uploaded in R studio and performs a logistic regression that calculates odds ratios, p-values and corrects for multiple testing. The additive genomic model was used assuming that each allele contributes a fixed amount of risk that is additive. Covariates such as sex, ethnicity, and age were incorporated into the analysis.

To infer the power of the analysis, the simulation study by Verma et al. (2018) [22] was used as a reference. Thus, the present study contains a total of 900 individuals in the group that received more anesthesia and 389 in the group that required less anesthetic, which reaches an approximate 1:3 ratio. When considering the total sample (n = 1,289) or the group that required more anesthetic for regular dental treatment (n = 900), there was enough power to detect associations if allele frequency was at least 5% and penetrance was at least 20%. For the group that required less anesthetic, there was enough power if allele frequency was 20% and penetrance was 20% or higher [22].

**Table 2. Characteristics of the selected SNPs.**

| SNP | Gene/Gene Product | Chromosome | Reference Allele | Polymorphic Allele | MAF | Consequence | Effect of pain / analgesia |
|---|---|---|---|---|---|---|---|
| **rs4818** | *COMT* Catechol-O-methyltransferase | 22 | C | G | 0.29 | Synonymous Variant | Intensity, sensitivity and duration of pain [11,12]. |
| **rs6269** | *COMT* Catechol-O-methyltransferase | 22 | A | G | 0.37 | Intron Variant | Pain sensitivity and opioid efficacy post-surgical pain [12,13]. |
| **rs3783641** | *GCH1* GTP cyclohydrolase 1 | 14 | T | A | 0.22 | Intron Variant | Modulates pain sensitivity [14]. |
| **rs6276** | *DRD2* Dopamine Receptor D2 | 11 | C | T | 0.46 | 3 Prime UTR Variant | Significant association with pain scores [15]. |
| **rs1799971** | *OPRM1* Opioid receptor mu 1 | 6 | A | G | 0.22 | Missense variant | Enhanced sensitivity for physical pain perception [16,17]. |
| **rs6795970** | *SCN10A* Sodium voltage-gated channel alpha subunit 10 | 3 | G | A | 0.25 | Missense variant | Modulates pain sensitivity [18]. Higher risk for inadequate analgesia [19]. |
| **rs6746030** | *SCN9A* Sodium voltage-gated channel alpha subunit 9 | 2 | G | A | 0.12 | Missense variant | Associated with variability of basal pain sensitivity [20]. |

In addition to the plots generated through the R package, the Web-based PheWAS-View was used to visually integrate the PheWAS results [23].

The comparison between the genotypic and allelic frequencies of the SNPs between groups was performed using the chi-square test implemented by R Studio program.

## Results

The results of the PheWAS analyses performed in the total sample (n = 1,289), in the group of individuals who received more anesthesia (n = 900) and in the group of individuals who received less anesthesia (n = 389) showed only nominal associations (p values between 0.00025 and 0.05) for nine, ten and five phenotypes, respectively (Table 3 and Fig 2). No associations survived the strict Bonferroni correction for these analyses.

When PheWAS was performed for each SNP separately using the total sample, the SNP rs1799971 of the *OPRM1* gene was nominally associated with five different phenotypes. Additionally, there was an association that survived Bonferroni correction (p<0.002) between *OPRM1* and asthma (phecode 495) as shown in Table 4 and Fig 3. This association was between the polymorphic allele G conferring a protective effect toward asthma.

Considering that pain can be influenced by factors such as sex and ethnicity, we performed a separate analysis for females and males, considering whether they received a greater or lesser amount of anesthesia. For ethnicity, a similar analysis was also performed considering all individuals who self-reported as White or Black and the amount of anesthesia received. The SNP rs3783641 in *GCH1* was associated with disorders of salivary secretion (phecode 527.7) when the PheWAS was performed for the group of all female subjects who received the highest amount of anesthesia (Table 4 and Fig 2).

The PheWAS performed in the group of self-reported Whites who received less anesthesia showed an association between the rs1799971 in *OPRM1* and gingival and periodontal diseases (phecode 523) as well as with periodontitis (acute or chronic) (phecode 523.3), the later providing a protective effect toward the phenotype (odds ratio = 0.17, 95% confidence interval 0.05–0.42, p = 0.0008) (Table 4 and Fig 2).

The variant rs4818 in *COMT* and its association with diseases and other conditions of the tongue (phecode 529) in the group consisting of self-reported Black individuals who received

**Table 3. Summary of the results of PheWAS analysis of p-values between 0.00025 and 0.05 for the total sample, for the group that received more anesthesia and for the group that received the least amount of anesthesia.**

| Phenotype | Phecode | SNP/Allele | Odds Ratio (95% confidence interval) | P-value | Affected | Non affected | Allele frequency |
|---|---|---|---|---|---|---|---|
| **Total sample** | | | | | | | |
| Disorders of tooth development | 520 | rs4818_G | 0.64 (0.43–0.94) | 0.02 | 71 | 1193 | 0.37 |
| Temporomandibular joint disorders | 526.4 | rs6276_T | 1.18 (1.01–1.39) | 0.04 | 523 | 742 | 0.63 |
| Disturbance of salivary secretion | 527.7 | rs3783641_A | 0.72 (0.53–0.97) | 0.03 | 187 | 1087 | 0.2 |
| Periodontitis (acute or chronic) | 523.3 | rs1799971_G | 0.69 (0.46–0.98) | 0.05 | 195 | 1081 | 0.13 |
| Asthma | 495 | rs1799971_G | 0.43 (0.26–0.67) | 0.0005 | 154 | 1118 | 0.13 |
| Spontaneous ecchymoses | 287.1 | rs1799971_G | 0.52 (0.26–0.93) | 0.04 | 86 | 1187 | 0.13 |
| Hypertension | 401 | rs1799971_G | 0.68 (-.48–0.94) | 0.02 | 302 | 971 | 0.13 |
| Complication of internal orthopedic device | 858 | rs1799971_G | 1.77 (0.99–3.01) | 0.04 | 50 | 1220 | 0.13 |
| Anomalies of jaw size/symmetry | 526.3 | rs6795970_A | 0.51 (0.27–0.89) | 0.02 | 34 | 1240 | 0.35 |
| **More anesthesia** | | | | | | | |
| Disorders of tooth development | 520 | rs4818_G | 0.62 (0.38–0.97) | 0.04 | 50 | 832 | 0.37 |
| Asthma | 495 | rs4818_G | 0.72 (0.53–0.97) | 0.03 | 111 | 770 | 0.37 |
| Dental caries | 521.1 | rs6276_T | 1.78 (1.16–2.78) | 0.009 | 841 | 38 | 0.38 |
| Temporomandibular joint disorders | 526.4 | rs6276_T | 1.22 (1.01–1.47) | 0.04 | 388 | 491 | 0.38 |
| Disturbance of salivary secretion | 527.7 | rs3783641_A | 0.64 (0.44–0.92) | 0.02 | 134 | 755 | 0.2 |
| Stomatitis and mucositis | 528.1 | rs3783641_A | 1.89 (0.99–3.48) | 0.04 | 23 | 866 | 0.2 |
| Asthma | 495 | rs1799971_G | 0.47 (0.26–0.79) | 0.007 | 112 | 777 | 0.12 |
| Spontaneous ecchymoses | 287.1 | rs1799971_G | 0.43 (0.18–0.89) | 0.04 | 63 | 827 | 0.12 |
| Stomatitis and mucositis | 528.1 | rs6795970_A | 0.41 (0.18–0.84) | 0.02 | 22 | 868 | 0.36 |
| Spontaneous ecchymoses | 287.1 | rs6795970_A | 1.54 (1.06–2.25) | 0.02 | 63 | 827 | 0.36 |
| **Less anesthesia** | | | | | | | |
| Other diseases of the teeth and supporting structures | 525 | rs4818_G | 1.49 (1.01–2.19) | 0.04 | 79 | 303 | 0.36 |
| Diseases of pulp and periapical tissues | 522 | rs6276_T | 1.39 (1.02–1.9) | 0.04 | 243 | 143 | 0.34 |
| Periodontitis (acute or chronic) | 523.3 | rs1799971_G | 0.48 (0.23–0.88) | 0.03 | 86 | 300 | 0.13 |
| Asthma | 495 | rs1799971_G | 0.30 (0.09–0.73) | 0.02 | 42 | 341 | 0.13 |
| Stomatitis and mucositis | 528.1 | rs6795970_A | 1.85 (1.09–3.15) | 0.02 | 31 | 353 | 0.34 |

less anesthesia is shown in Table 4 and Fig 2. In this analysis, it is observed that the SNP rs4818 confers a five-time increase in risk for the phenotype (odds ratio = 5.07, 95% confidence interval 1.87–15.96, p = 0.002).

The genotypic and allelic distributions for all SNPs studied according to the study groups are shown in Appendix Table 1. We observed statistically significant differences in the genotypic and allelic frequencies for the rs6276 in *DRD2*. For the rs1799971 in *OPRM1*, there was a difference between allelic frequencies between groups that received a greater or lesser amount of anesthesia. Furthermore, it can be observed that the C allele (wild) of the SNP rs6276 was associated with the group that received the highest amount of anesthesia with a risk of 1.2 (95% confidence interval 1.01–1.44, p = 0.03). While the G allele of the SNP rs1799971 confers a protective effect on the group that received the least amount of anesthesia (odds ratio = 0.62, 95% confidence interval 0.41–0.95, p = 0.02).

## Discussion

Our results demonstrated an association between the rs1799971 in *OPRM1* and asthma. This finding is consistent with the results from a recent study that assessed asthma severity in

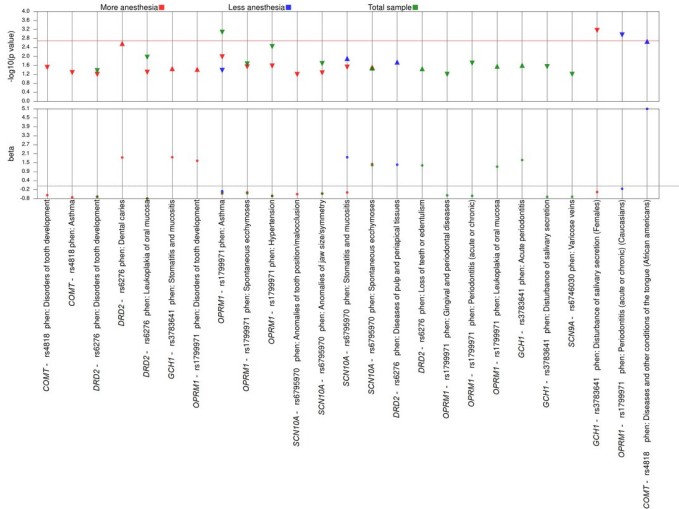

**Fig 2. PheWAS results stratified across different groups.** The color red represents a group of individuals that received more anesthesia. Blue represents a group that received less anesthesia, and green represents the total sample. The tip direction of the triangles represents the direction of the odds ratio of each association, the upward triangles indicate OR ≥ 1; downward triangles indicate a protective effect (OR < 1.0). Nominal associations are showed under the horizontal red line that indicates threshold of p < 0.002.

outpatients carrying rs1799971 [24]. In this study the authors also used an experimental asthma rodent model harboring the functionally equivalent SNP to investigate the mechanism by which this SNP influences the allergic immune response. They found that patients with asthma carrying the *OPRM1* GG genotype exhibited enhanced airway hyperresponsiveness, attributable to enhanced Th2 cell differentiation in the regional lymph node. In our study we found not only asthma associated with the allele G (polymorphic) of rs1799971, but also that

**Table 4. Summary of the PheWAS results of p-values below 0.05 for each SNP separately, using the total sample, and the groups that received more and less anesthesia, considering sex and ethnicity.** * Indicates statistically significant results after correction for multiple testing.

| Phenotype | Phecode | SNP/Allele | Odds Ratio (95% Confidence Interval) | P-value | Affected | Non affected | Allele frequency |
|---|---|---|---|---|---|---|---|
| **Total sample** | | | | | | | |
| Gingival and periodontal diseases | 523 | rs1799971_G | 0.61 (0.35–0.97) | 0.05 | 105 | 1171 | 0.13 |
| Periodontitis (acute or chronic) | 523.3 | rs1799971_G | 0.64 (0.44–0.9) | 0.01 | 195 | 1081 | 0.13 |
| Leukoplakia of oral mucosa | 528.6 | rs1799971_G | 1.28 (1.02–1.61) | 0.03 | 467 | 809 | 0.13 |
| Asthma | 495 | rs1799971_G | 0.44 (0.27–0.69) | 0.0006* | 154 | 1118 | 0.13 |
| Spontaneous ecchymoses | 287.1 | rs1799971_G | 0.47 (0.24–0.83) | 0.02 | 86 | 1187 | 0.13 |
| Hypertension | 401 | rs1799971_G | 0.63 (0.46–0.85) | 0.003 | 302 | 971 | 0.13 |
| **Females—more anesthesia** | | | | | | | |
| Gingival and periodontal diseases | 523 | rs3783641_A | 1.82 (1.08–3.02) | 0.02 | 40 | 465 | 0.21 |
| Chronic periodontitis | 523.32 | rs3783641_A | 0.18 (0.03–0.61) | 0.02 | 20 | 485 | 0.21 |
| Disturbance of salivary secretion | 527.7 | rs3783641_A | 0.39 (0.22–0.64) | 0.0005* | 80 | 425 | 0.21 |
| **Whites—less anesthesia** | | | | | | | |
| Gingival and periodontal diseases | 523 | rs1799971_G | 0.22 (0.04–0.71) | 0.04 | 32 | 262 | 0.12 |
| Periodontitis (acute or chronic) | 523.3 | rs1799971_G | 0.17 (0.05–0.43) | 0.0008* | 69 | 225 | 0.12 |
| **Blacks—less anesthesia** | | | | | | | |
| Diseases and other conditions of the tongue | 529 | rs4818_G | 5.08 (1.87–15.97) | 0.002* | 37 | 33 | 0.2 |

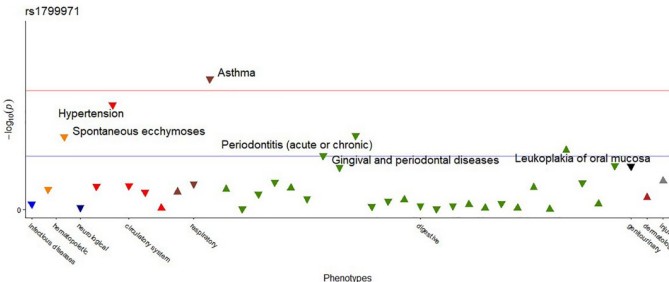

**Fig 3. PheWAS result for SNP rs1799971 performed with the total sample.** The blue line indicates p-value <0.05. The horizontal red line indicates the threshold of p < 0.002. The tip direction of the triangles represents the direction of the odds ratio of each association, the upward triangles indicate odds ratio≥1; downward triangles indicate a protective effect (odds ratio<1.0). The different colors of the triangles indicate different disease groups.

patients that carry this allele needed less amounts of anesthesia compared to patients who carried the reference allele for the same dental procedures.

Among the salivary secretion disorders, the prevalence rates for sialorrhea for the general population are unknown, while xerostomia is more frequent in women [25]. The SNP 3783641 of the *GCH1* gene showed a protective effect for disturbance of salivary secretion in the group of women who received more anesthesia. Since rs3783641 is an intronic variant, no clinical significance has been reported in the literature. However, a higher pain tolerance has been attributed to a haplotype including the rs3783641 in the *GCH1* gene described by [14]. Although a haplotype in *GCH1* may be constructed with 15 SNPs [26], 100% sensitivity and specificity can be obtained through the screening of only three variants in untranslated regions, including the rs3783641. In summary, most clinical studies investigating this *GCH1* haplotype have confirmed its association with higher pain tolerance [27]. On the other hand, studies investigating rs3783641 are less conclusive [28,29]. *GCH1* plays an important role in the biosynthesis of the cofactor Bh4, which is essential for the synthesis of catecholamines, such as norepinephrine, that participate in the neuronal regulation of salivary secretion through its binding with the β-adrenergic receptors.

A protective effect of rs1799971 toward having periodontitis was demonstrated in the group of self-reported Whites who received less anesthesia. Periodontitis is more prevalent in certain ethnic populations, especially non-Hispanic Black individuals. Literature also indicates that individuals of African descent are more prone to periodontitis [30,31]. Although we understand that periodontitis is also influenced by the environment [32], data from different populations in the 1000 Genome show that the frequency of the G allele (polymorphic) for the African population is lower (1%), followed by Europeans (16%), Americans (20%), East Asians (39%), and South Asians (42%). Hence, it is unlikely we had enough power to detect a possible association between this variant and periodontitis in self-reported Black individuals. A mechanism by which this functional SNP could explain the protective effect toward having periodontitis would be through the higher affinity for beta-endorphin, which participates in regulating the secretion of pro-inflammatory cytokines from periphery immune cells [33]. Furthermore, immune response and inflammation pathways, such as those involving cytokine and chemokine activities, have been reported to be the most upregulated in RNA expression analysis of affected tissues from individuals with periodontitis [34]. Additionally, carriers of the G allele had lower serum concentrations of cytokines interleukin-6 (IL-6), tumor necrosis factor-α (TNF-α), and interferon-γ (IFN-γ), and showed a higher quality of life compared to individuals without this allele [35].

Ethnic differences in oral health and orofacial pain have been reported in the literature. Therefore, the association of the rs4818 SNP with disease of the tongue in the African-American group that received less anesthesia should be considered with caution as it may be linked to other disparities, as ethnic groups vary in lifestyle, socioeconomic conditions and access to dental services [8,36]. The rs4818 G allele (polymorphic) was reported to be protective for oral pain [11]. The same allele has been also associated with the risk of developing one of the most common diseases of the tongue, known as benign migratory glossitis, with a risk 4.7 times greater for developing the condition in individuals with the GC genotype [37].

Although a direct relationship between *COMT* and the phenotype of tongue disease has not been reported to date, the catechol-methyltransferase enzyme encoded by this gene acts on the degradation of catecholamines that play important roles in several physiological processes and consequently also in diseases. As an example, the dopamine responsive burning mouth syndrome, that is characterized by a burning sensation of the oral mucosa in the absence of clinically apparent mucosal changes, mainly affects the tongue [38].

In spite of rs6276 (*DRD2*) had only nominative associations in the PheWAS analysis, the genotypic and allelic frequencies were significantly different between groups of individuals separated by the amount of anesthetic needed for regular dental procedures (p = 0.02) and the C allele (wild) was associated with the risk of needing more anesthesia (odds ratio = 1.2, 95% confidence interval 1.01–1.49; p = 0.03). These results corroborate the study by Qadri et al. (2015) [15], who showed that individuals with two copies of this allele had higher pain scores and also the same allele was positively associated with both opioid analgesic use and pain severity in emergency departments (p = 0.03). The rs6276 occurs in the 3′UTR region, which is important for the stability and regulation of the D2R dopamine receptor mRNA [39].

Lidocaine, articaine, and mepivacaine are the three most used anaesthetics in dentistry. Mepivacaine has the same potency s lidocaine, with milder vasodilating ability. The duration of effect in a maxillary infiltration of 2% lidocaine with 1:100,000 epinephrine ranges from 60 to 90 minutes in pulp tissue and 170 to 300 minutes in soft tissue, while the duration of 4% articaine with 1:100,000 epinephrine encompasses 60 to 75 minutes in the pulp and 170 to 360 minutes in the soft tissues. To mention another circumstance, palate anesthesia is unpleasant for many patients because it is a very sensitive site, so when the anesthetic used is articaine, it is not necessary to perform infiltration in the palatal location for extraction procedures, as it is very effective only with injection in the region maxillary vestibular. Articaine presents tissue diffusion when compared to the lidocaine in areas other than the place where the anesthetic was deposited, avoiding excessive punctures and patient discomfort [40–44]. Therefore, depending on the dental procedure, the type of anesthetic could influence the number of tubetes to be used. To minimize the potential impact of these differences in our study, we matched individuals by the treatment procedure they needed, but still some residual effect of these differences could have influenced the results.

The inclusion of children could be seen as a limitation, since we use weight calculate the maximum amount of anaesthetic to be used for their treatment. In this way, children necessarily must use less tubetes then adults [45]. Since the parameter of "pain perception" was the amount of anesthetic used, these variables involved in metabolization, duration effect, toxicity and others, could impact the interpretation of the results. However, only five children under 12 years of age were included in the study (ages 8, 10.3, 10.6. 11.3, and 11.5 years). The potential impact of these subjects in the results presented here is likely very small or neglectable.

Possible additional limitations of the present study are the lack of a pain measurement model that would allow assessing the pain threshold among participants according to the amount of anesthesia they received. Second, the use of self-reported ethnicity may in certain cases not actually correspond to ancestry, where some self-identified African Americans may

have European ancestry and some self-identified European Americans have substantial admixture from African ancestry [46]. Finally, the data studied come from a registry, which may include undetected errors and variation of clinical descriptions.

Nevertheless, our study provides a pioneering approach to the study of pain perception by using "reverse genetics" through the PheWAS methodology (a study design in which the association between single nucleotide polymorphisms or other types of DNA variants is tested across many different phenotypes), considered an unbiased approach to test for associations between a specific genetic variant or combination of variants, and a wide range of phenotypes [47]. It was possible to identify new associations between SNPs in genes related to pain and oral phenotypes in the groups of individuals who received more anesthesia or who received less anesthesia. We are careful to not directly equate receiving anesthesia to feeling more or less pain, since the amount of anesthetic used can be influenced by several factors, such as anatomical variations, metabolism rate, type of anesthetic used, local tissue pH (which can be modified by the presence of inflammation), technique used, and not just pain.

These associations bring promising opportunities to better understand not only pain perception but also the phenotypes that we found associated, such as asthma, disturbance of salivary secretion, periodontitis, and diseases affecting the tongue. Additionally, these results allow for new perspectives regarding the relationship between important pathways related to dopamine (*DRD2*, *COMT*, and *GCH1*) and immune system response pathways (*OPRM1*). More importantly, we hope that due to the association with risk of needing more anesthesia found in the present study, the SNPs rs6276_C and rs1799971_A are potential candidates to be studied as pain biomarkers, a priori for dental treatment in the future.

## Supporting information

**S1 Data.**
(XLSX)

## Acknowledgments

We would like to acknowledge the Dental Registry and DNA Repository project, which is supported by the University of Pittsburgh School of Dental Medicine. Also, we thank the Graduate Program in Oncology and Medical Sciences at the Federal University of Pará, Brazil, and the National Center for Advancing Translational Sciences of the National Institutes of Health, for the support to Lisboa, RO, and Bezamat, M, respectively. This manuscript would not be possible without their contributions.

## Author Contributions

**Conceptualization:** Raymond F. Sekula, Luiz Carlos Santana-da-Silva, Alexandre R. Vieira.

**Data curation:** Mariana Bezamat, Kathleen Deeley, Alexandre R. Vieira.

**Formal analysis:** Rosany O. Lisboa, Mariana Bezamat, Kathleen Deeley, Alexandre R. Vieira.

**Funding acquisition:** Raymond F. Sekula, Luiz Carlos Santana-da-Silva, Alexandre R. Vieira.

**Investigation:** Raymond F. Sekula, Alexandre R. Vieira.

**Methodology:** Rosany O. Lisboa, Mariana Bezamat, Kathleen Deeley, Alexandre R. Vieira.

**Project administration:** Kathleen Deeley, Alexandre R. Vieira.

**Resources:** Alexandre R. Vieira.

**Supervision:** Luiz Carlos Santana-da-Silva, Alexandre R. Vieira.

**Validation:** Rosany O. Lisboa.

**Visualization:** Rosany O. Lisboa, Alexandre R. Vieira.

**Writing – original draft:** Rosany O. Lisboa.

**Writing – review & editing:** Raymond F. Sekula, Mariana Bezamat, Kathleen Deeley, Luiz Carlos Santana-da-Silva, Alexandre R. Vieira.

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
