## [Decision Letter · Decision Letter 0]

4 Jul 2022

PONE-D-22-12279Pain Perception Genes, Asthma, and Oral Health: A Reverse Genetics StudyPLOS ONE

Dear Dr. Vieira,

Thank you for submitting your manuscript to PLOS ONE. After careful consideration, we feel that it has merit but does not fully meet PLOS ONE’s publication criteria as it currently stands. Therefore, we invite you to submit a revised version of the manuscript that addresses the points raised during the review process.

 There are significant request to elaborate on experimental design, subject and target selection, and the novel approach used by the grup.

We look forward to receiving your revised manuscript.

Kind regards,

JJ Cray Jr., Ph.D.

Academic Editor

PLOS ONE

Journal Requirements:

"We would like to acknowledge the Dental Registry and DNA Repository project, which is supported by the University of Pittsburgh School of Dental Medicine. Also, we thank the Graduate Program in Oncology and Medical Sciences at the Federal University of Pará, Brazil, and the National Center for Advancing Translational Sciences of the National Institutes of Health, for the support to Lisboa, RO, and Bezamat, M, respectively.”

Reviewers' comments:

Reviewer's Responses to Questions

**Comments to the Author**

1. Is the manuscript technically sound, and do the data support the conclusions?

Reviewer #1: No

Reviewer #2: Yes

2. Has the statistical analysis been performed appropriately and rigorously? 

Reviewer #1: Yes

Reviewer #2: Yes

3. Have the authors made all data underlying the findings in their manuscript fully available?

Reviewer #1: Yes

Reviewer #2: No

4. Is the manuscript presented in an intelligible fashion and written in standard English?

Reviewer #1: Yes

Reviewer #2: Yes

5. Review Comments to the Author

Reviewer #1: 1. Was only one type of anesthetic compared? Example: lidocaine with vasoconstrictor. And in similar clinical situations? Example: in third molar extractions or class I restorations or endodontic treatment, etc. It is necessary to describe what types of dental treatment were used as a parameter in anesthesia, and what anesthetics were used.

2. In table 01, include what the numbers inside the parentheses mean (maximum and minimum?). If so, it is important to distinguish adults from children, since the parameter used to evaluate the groups was the amount of anesthetic during the dental procedure; in children, fear and anxiety greatly influence pain, especially during procedures involving needles, as is the case with anesthesia.

3. Was it the same person who anesthetized all participants? Has any calibration been done between dentists? What kind of treatment was done on everyone to compare anesthesia?

4. The presence of some inflamed tissue can influence the amount of anesthetic tubes to be used in each procedure, was this taken into account?

5. How were these phenotypes evaluated? More detail is needed.

6. It is necessary to mention which studies were used as a reference in the choice of SNPs.

7. It is necessary to include the background of the selected SNPs in the introduction.

8. Add in table 02 information such as “increase or decrease pain sensitivity” other than just “Intensity or pain sensitivity”. So we would know what exactly the polymorphism effect on pain perception.

9. Was an epistasis analysis performed? Some studies demonstrate that the GCH1 gene can modulate the effect of COMT on the pain response. Patients who have SNP in both genes usually have a protective effect (analgesia) by GCH1, even with the COMT SNP still present.

10. I suggest increasing the quality of the images, it is difficult to understand what is written.

11. Results need to be organized more clearly, it's confusing.

12. I suggest further specifying the purpose of the work. Through the methodology of the work, the objective was to investigate these clinical phenotypes and some genetic markers in the response/efficacy of anesthetic or analgesia and not necessarily pain. It is necessary to clarify the justification and background of these objectives during the introduction, as it is not clear exactly what this work is about, only during the reading of the methodology. There are numerous ways to work with or assess pain: acute, spontaneous, chronic, stimulated/induced. It is necessary to specify and better describe the type of pain being investigated in this study, since one of the objectives is precisely to make this correlation between pain-related SNPs and other clinical phenotypes.

13. Importantly, the SNPs investigated here are not exclusively related to nociceptive pathways, but also to other non-painful conditions. It is necessary to describe each one of them better in this sense, depending on the amount of clinical phenotypes that were included in the analysis and are not directly related to nociceptive processes. Furthermore, it is necessary to include this observation in the discussion of the work.

14. The title is a few confusing: “Pain Perception Genes, Asthma, and Oral Health: A Reverse Genetics Study”. What does it mean “reverse genetics study” ?

15. The only relationship with pain in this work was the way the sample was divided into two groups – and this division, and the genes that are really related to painful processes, still need to be more detailed. However, the immense amount of clinical phenotypes evaluated are not directly related to pain, it is necessary to establish a link between these phenomena and describe them very well in the study.

16. It should also be discussed that there was not found any difference between the both groups after Bonferroni correction. Perhaps the method of using the amount of anesthetic as a parameter for dividing the groups was not satisfactory for evaluating these clinical phenotypes (pain-related? What exactly?) with genetic markers of pain. There are many factors that influence the amount of anesthetic used. This was a decisive criterion in the selection of the two groups, however, the way in which it was made was neither mentioned nor described. If there was calibration between the dentists who performed the procedure (both anesthetic and dental – surgical/restorative/endodontic/periodontal – which should also be described), nor if there was any nervous branch of predilection and region that was anesthetized – maxilla or mandible, if there was previous pain or local inflammation/infection. These are all factors that influence the anesthesia procedure, and that influenced the choice/division of the study sample.

17. The final considerations state that “Nevertheless, our study provides a pioneering approach to the study of pain perception”. What tool was used to evaluate pain perception? Visual analogue scale (VAS)? Sound, eye, and motor (SEM) indexes? Quantitative Sensory Testing?

18. Which previous study was used to determine the cut-off point for the amount of anesthetic? So it could be used as parameter to dived the sample into two groups.

Reviewer #2: The authors performed PheWAS-like analysis to examine the associations of 58 phenotypes with 7 pain related SNPs. A total of 1289 subjects were selected and grouped based on the dose of anesthesia received during the dental treatment procedure. They found an association between the rs1799971 in OPRM1 and asthma, which was in agreement with a recent report. In addition, the study identified three associations of oral phenotypes with variants of pain-related genes. Overall, the study is carefully designed and performed.

Major critiques:

1. The authors applied novel criteria to recruit subjects - individuals who feel more pain (received more anesthesia) or less pain (who received less anesthesia). However, using “anesthesia utilization” as a surrogate indicator of pain severity raised concerns. It is unknown if other factors were taken into consideration, e.g. medication history, especially opioids and other analgesic drugs, dental procedure types (location, duration), as well as pain assessment (score, type). Whether these variants were adjusted in the statistical analysis was not mentioned in the manuscript.

2. The sample size of "black – less anesthesia" group is small. The authors should address how they controlled the power of analysis to ensure the association detected between rs4818 of COMT and diseases of the tongue was not false-positive.

3. The authors should address the rationale of selecting these 7 SNPs, in terms of their relationships with orofacial pain perception, specifically.

Minor issues:

Line 233, Appendix Table 1 is not included in the manuscript.

Line 143, reference is missing.

6. PLOS authors have the option to publish the peer review history of their article (what does this mean?). If published, this will include your full peer review and any attached files.

Reviewer #1: **Yes: **Flávia Fonseca Carvalho Soares

Reviewer #2: No

---

## [Author Response · Author response to Decision Letter 0]

11 Aug 2022

Journal Requirements:

All information has been revised as requested.

The additional information in the acknowledgment section was excluded. It is marked in yellow and crossed out with a line.

"We would like to acknowledge the Dental Registry and DNA Repository project, which is supported by the University of Pittsburgh School of Dental Medicine. Also, we thank the Graduate Program in Oncology and Medical Sciences at the Federal University of Pará, Brazil, and the National Center for Advancing Translational Sciences of the National Institutes of Health, for the support to Lisboa, RO, and Bezamat, M, respectively.”

The additional information in the acknowledgment section was excluded. It is marked in yellow and crossed out with a line. We added it in the cover letter as asked.

We added a supporting information file with our raw data.

We changed the cover letter to reflect that we are uploading all raw data as an appendix.

Reviewers' comments:

Reviewer #1:

1. Was only one type of anesthetic compared? Example: lidocaine with vasoconstrictor. And in similar clinical situations? Example: in third molar extractions or class I restorations or endodontic treatment, etc. It is necessary to describe what types of dental treatment were used as a parameter in anesthesia, and what anesthetics were used.

The information of the local anesthetics used were added to the manuscript in the methods section (line 134) and also the duration of anesthesia according to the article reference.

2. In table 01, include what the numbers inside the parentheses mean (maximum and minimum?). If so, it is important to distinguish adults from children, since the parameter used to evaluate the groups was the amount of anesthetic during the dental procedure; in children, fear and anxiety greatly influence pain, especially during procedures involving needles, as is the case with anesthesia.

The number inside the parentheses mean the range of age in each group. As you can see in the table 1 the groups were matched for age, so there are no significant variations between groups. Both children and adults may display some fear and anxiety. 90% of pediatric dentistry is done under local anesthesia. Comparison groups were originally matched by age to mitigate the influence of age.

3. Was it the same person who anesthetized all participants? Has any calibration been done between dentists? What kind of treatment was done on everyone to compare anesthesia?

Data for the study comes from our registry, that currently spans over 17 years. It was not the same operator that anesthetized all participants. All dentists in training perform anesthesia under the same protocol. Treatments done were typical operative dentistry procedures. We added these clarifications in the Methods section (line 137) to address this concern.

4. The presence of some inflamed tissue can influence the amount of anesthetic tubes to be used in each procedure, was this taken into account?

To mitigate this aspect, comparison groups were matched by dental procedure. But beyond that, it was not possible to fully account for levels of inflammation at the day treatments were performed. We added a clarification in the method section (line 139) to acknowledge this detail.

5. How were these phenotypes evaluated? More detail is needed.

Phenotypes were defined based on coding used in the registry, which reflects descriptions at the dental records. We added this information in the methods section (line 159) to address this concern.

6. It is necessary to mention which studies were used as a reference in the choice of SNPs.

The studies used as references for each SNP were originally mentioned in table 2.

7. It is necessary to include the background of the selected SNPs in the introduction.

The background of the SNPs was inserted in the introduction.

8. Add in table 02 information such as “increase or decrease pain sensitivity” other than just “Intensity or pain sensitivity”. So we would know what exactly the polymorphism effect on pain perception.

 The SNPs in the chosen genes act mainly in the general context of the pain perception process and unfortunately in the literature there are no specific concrete associations of these SNPs regarding their participation in the increase or decrease of pain, many of them seem to act in both situations depending on some factors, therefore, to avoid future mistakes when conditioning them with a specific association, it is preferable to use the term “Intensity or pain sensitivity” as used in the reference articles. Among the suggested explanations for the discrepancy found in the literature is that this may indicate that these polymorphisms affect pain sensations of some, but not all, stimuli. This information will be added to the manuscript to explain why we used the term.

9. Was an epistasis analysis performed? Some studies demonstrate that the GCH1 gene can modulate the effect of COMT on the pain response. Patients who have SNP in both genes usually have a protective effect (analgesia) by GCH1, even with the COMT SNP still present.

 No. For this specific paper epistasis analysis was not performed because the study's main objective was to search associations using the PheWAS methodology and not the influence that could exist between the studied markers. 

10. I suggest increasing the quality of the images, it is difficult to understand what is written.

All images were submitted to the PACE tool and the quality of the images was enhanced.

11. Results need to be organized more clearly, it's confusing.

The organization was done according to the specific results of each PheWAS analysis performed and following the presentation standard for this methodology.

12. I suggest further specifying the purpose of the work. Through the methodology of the work, the objective was to investigate these clinical phenotypes and some genetic markers in the response/efficacy of anesthetic or analgesia and not necessarily pain. It is necessary to clarify the justification and background of these objectives during the introduction, as it is not clear exactly what this work is about, only during the reading of the methodology. There are numerous ways to work with or assess pain: acute, spontaneous, chronic, stimulated/induced. It is necessary to specify and better describe the type of pain being investigated in this study, since one of the objectives is precisely to make this correlation between pain-related SNPs and other clinical phenotypes.

The information in the main sections of the manuscript has been improved for a better understanding of the purpose of the study and how it was done.

13. Importantly, the SNPs investigated here are not exclusively related to nociceptive pathways, but also to other non-painful conditions. It is necessary to describe each one of them better in this sense, depending on the amount of clinical phenotypes that were included in the analysis and are not directly related to nociceptive processes. Furthermore, it is necessary to include this observation in the discussion of the work.

Most of phenotypes included in the analysis have a pain sensitivity component. SNPs were chosen because they have some association with the pain perception processor regardless of whether it is nociceptive pain or not. And since the perception of pain is something extremely complex with the involvement of several genes, it is expected that these SNPs participate in other processes and are also associated with other non-painful conditions.

14. The title is a few confusing: “Pain Perception Genes, Asthma, and Oral Health: A Reverse Genetics Study”. What does it mean “reverse genetics study” ?

In reverse genetics, genetic marker data are used to drive, or form the basis of, new phenotype definitions, that is, the analysis starts from genotype to phenotype. We addeda clarification in the Introduction section (line 83)/

15. The only relationship with pain in this work was the way the sample was divided into two groups – and this division, and the genes that are really related to painful processes, still need to be more detailed. However, the immense amount of clinical phenotypes evaluated are not directly related to pain, it is necessary to establish a link between these phenomena and describe them very well in the study.

Most phenotypes included actually have potentially a pain sensitivity component associated with them. We added this clarification in the methods section (line 162).

16. It should also be discussed that there was not found any difference between the both groups after Bonferroni correction. Perhaps the method of using the amount of anesthetic as a parameter for dividing the groups was not satisfactory for evaluating these clinical phenotypes (pain-related? What exactly?) with genetic markers of pain. There are many factors that influence the amount of anesthetic used. This was a decisive criterion in the selection of the two groups, however, the way in which it was made was neither mentioned nor described. If there was calibration between the dentists who performed the procedure (both anesthetic and dental – surgical/restorative/endodontic/periodontal – which should also be described), nor if there was any nervous branch of predilection and region that was anesthetized – maxilla or mandible, if there was previous pain or local inflammation/infection. These are all factors that influence the anesthesia procedure, and that influenced the choice/division of the study sample.

Bonferroni is quite a strict method of multiple comparion correction. One result survived this correction. Results that do not survive a Bonferroni correction were also presented and have been added to the Discussion section, since they may represent true biological relationships and deserve further study. The reviewer's other considerations were supplied in the previous comments.

17. The final considerations state that “Nevertheless, our study provides a pioneering approach to the study of pain perception”. What tool was used to evaluate pain perception? Visual analogue scale (VAS)? Sound, eye, and motor (SEM) indexes? Quantitative Sensory Testing?

The final consideration was re-elaborated to suit the proposed objectives and results of the study.

18. Which previous study was used to determine the cut-off point for the amount of anesthetic? So it could be used as parameter to dived the sample into two groups.

It is common clinical practice to give up to one tubete of anesthetic for dental treatments. Hence, the cutoff used.

Reviewer #2: The authors performed PheWAS-like analysis to examine the associations of 58 phenotypes with 7 pain related SNPs. A total of 1289 subjects were selected and grouped based on the dose of anesthesia received during the dental treatment procedure. They found an association between the rs1799971 in OPRM1 and asthma, which was in agreement with a recent report. In addition, the study identified three associations of oral phenotypes with variants of pain-related genes. Overall, the study is carefully designed and performed.

Major critiques:

1. The authors applied novel criteria to recruit subjects - individuals who feel more pain (received more anesthesia) or less pain (who received less anesthesia). However, using “anesthesia utilization” as a surrogate indicator of pain severity raised concerns. It is unknown if other factors were taken into consideration, e.g. medication history, especially opioids and other analgesic drugs, dental procedure types (location, duration), as well as pain assessment (score, type). Whether these variants were adjusted in the statistical analysis was not mentioned in the manuscript.

Groups were matched by dental procedures, but chronic use of opioids or other analgesic oral drugs, was not considered. No pain assessments were performed. 

2. The sample size of "black – less anesthesia" group is small. The authors should address how they controlled the power of analysis to ensure the association detected between rs4818 of COMT and diseases of the tongue was not false-positive.

Test power was controlled using the simulation study by Verma et al. (2018) and according to this study for the group that required less anesthetic, there was sufficient power if the allele frequency was 20% and the penetrance was 20% or higher.

3. The authors should address the rationale of selecting these 7 SNPs, in terms of their relationships with orofacial pain perception, specifically.

SNPs were selected for their involvement with the perception of the general participation of genes in fundamental processes. Those SNPs that are related to orofacial pain were described in the introduction for better understanding. Per reviewer 1, rationale was added to the introduction section.

Minor issues:

Line 233, Appendix Table 1 is not included in the manuscript.

The appendix table1 was inserted in the main text as table 5.

Line 143, reference is missing.

The missing reference was added.

---

## [Decision Letter · Decision Letter 1]

9 Sep 2022

PONE-D-22-12279R1Pain Perception Genes, Asthma, and Oral Health: A Reverse Genetics StudyPLOS ONE

Dear Dr. Vieira,

Thank you for submitting your manuscript to PLOS ONE. After careful consideration, we feel that it has merit but does not fully meet PLOS ONE’s publication criteria as it currently stands. Therefore, we invite you to submit a revised version of the manuscript that addresses the points raised during the review process.

There are some outstanding issues mostly with interpretation that should be addressed in a revision.

We look forward to receiving your revised manuscript.

Kind regards,

JJ Cray Jr., Ph.D.

Academic Editor

PLOS ONE

Journal Requirements:

Reviewers' comments:

Reviewer's Responses to Questions

**Comments to the Author**

1. If the authors have adequately addressed your comments raised in a previous round of review and you feel that this manuscript is now acceptable for publication, you may indicate that here to bypass the “Comments to the Author” section, enter your conflict of interest statement in the “Confidential to Editor” section, and submit your "Accept" recommendation.

Reviewer #1: All comments have been addressed

Reviewer #2: All comments have been addressed

2. Is the manuscript technically sound, and do the data support the conclusions?

Reviewer #1: Partly

Reviewer #2: Yes

3. Has the statistical analysis been performed appropriately and rigorously? 

Reviewer #1: Yes

Reviewer #2: Yes

4. Have the authors made all data underlying the findings in their manuscript fully available?

Reviewer #1: Yes

Reviewer #2: Yes

5. Is the manuscript presented in an intelligible fashion and written in standard English?

Reviewer #1: Yes

Reviewer #2: Yes

6. Review Comments to the Author

Reviewer #1: The authors answered most of the questions raised and significantly improved the quality of the manuscript. However, there are still major concerns that need to be addressed.

1. Lidocaine and articaine have very different particularities. For instance, The duration of effect in a maxillary infiltration of 2% lidocaine with 1:100,000 epinephrine ranges from 60 to 90 minutes in pulp tissue and 170 to 300 minutes in soft tissue, while the duration of 4% articaine with 1:100,000 epinephrine encompasses 60 to 75 minutes in the pulp and 170 to 360 minutes in the soft tissues. Palate anesthesia is unpleasant for many patients because it is a very sensitive site, so when the anesthetic used is articaine, it is not necessary to perform infiltration in the palatal location for extraction procedures, as it is very effective only with injection in the region maxillary vestibular. Articaine presents tissue diffusion when compared to the lidocaine in areas other than the place where the anesthetic was deposited, avoiding excessive punctures and patient discomfort. Therefore, depending on the dental procedure, the type of anesthetic could influence the number of tubetes to be used. This should have been considered in the methodological strategy of the study.

2. Regarding including children in the sample, I still have concerns about the parameter of “quantity of anesthetic used”. Since we use weight calculate the maximum amount of anesthetic. In this way, children necessarily must use less tubetes then adults. Since the parameter of “pain perception” is the amount of anesthetic used, these variables involved in metabolization, duration effect, toxicity and others, should be addressed accordingly.

3. The final consideration state that “Through this powerful strategy, it was possible to identify new associations between SNPs in genes related to pain and oral phenotypes in the groups of individuals who feel more pain (those who received more anesthesia) or less pain (who received less anesthesia)”. Actually, it is not possible to establish this type of equivalence. Since the amount of anesthetic used can be influenced by several factors, such as anatomical variations, metabolism rate, type of anesthetic used, local tissue pH (which can be modified by the presence of inflammation), technique used, among others, and not just pain.

Reviewer #2: The authors have carefully addressed the questions from the reviewers. The revised manuscript is in better shape. However, there are couple places need additional inputs.

First of all, based on the analysis scale (58 phenotypes, 7 SNPs, 1289 subjects), it can not be called as a real PheWAS study. The authors should be careful when using the term in the manuscript.

Second, the authors should revise the Introduction to be more concise and focused. This is not a review paper.

7. PLOS authors have the option to publish the peer review history of their article (what does this mean?). If published, this will include your full peer review and any attached files.

Reviewer #1: **Yes: **Flávia Fonseca Carvalho Soares

Reviewer #2: No

---

## [Author Response · Author response to Decision Letter 1]

19 Sep 2022

Response to Reviews

Reviewer #1: The authors answered most of the questions raised and significantly improved the quality of the manuscript. However, there are still major concerns that need to be addressed.

1. Lidocaine and articaine have very different particularities. For instance, The duration of effect in a maxillary infiltration of 2% lidocaine with 1:100,000 epinephrine ranges from 60 to 90 minutes in pulp tissue and 170 to 300 minutes in soft tissue, while the duration of 4% articaine with 1:100,000 epinephrine encompasses 60 to 75 minutes in the pulp and 170 to 360 minutes in the soft tissues. Palate anesthesia is unpleasant for many patients because it is a very sensitive site, so when the anesthetic used is articaine, it is not necessary to perform infiltration in the palatal location for extraction procedures, as it is very effective only with injection in the region maxillary vestibular. Articaine presents tissue diffusion when compared to the lidocaine in areas other than the place where the anesthetic was deposited, avoiding excessive punctures and patient discomfort. Therefore, depending on the dental procedure, the type of anesthetic could influence the number of tubetes to be used. This should have been considered in the methodological strategy of the study.

RESPONSE: We added discussion to acknowledge this limitation.

2. Regarding including children in the sample, I still have concerns about the parameter of “quantity of anesthetic used”. Since we use weight calculate the maximum amount of anesthetic. In this way, children necessarily must use less tubetes then adults. Since the parameter of “pain perception” is the amount of anesthetic used, these variables involved in metabolization, duration effect, toxicity and others, should be addressed accordingly.

RESPONSE: We added discussion and acknowledged this limitation.

3. The final consideration state that “Through this powerful strategy, it was possible to identify new associations between SNPs in genes related to pain and oral phenotypes in the groups of individuals who feel more pain (those who received more anesthesia) or less pain (who received less anesthesia)”. Actually, it is not possible to establish this type of equivalence. Since the amount of anesthetic used can be influenced by several factors, such as anatomical variations, metabolism rate, type of anesthetic used, local tissue pH (which can be modified by the presence of inflammation), technique used, among others, and not just pain.

RESPONSE: We rewrote the statement to address this concern.

Reviewer #2: The authors have carefully addressed the questions from the reviewers. The revised manuscript is in better shape. However, there are couple places need additional inputs.

First of all, based on the analysis scale (58 phenotypes, 7 SNPs, 1289 subjects), it can not be called as a real PheWAS study. The authors should be careful when using the term in the manuscript.

RESPONSE: The definition of PheWAS is a study design in which the association between single nucleotide polymorphisms or other types of DNA variants is tested across a large number of different phenotypes. Based on the definition, our study is a PheWAS study. We added this definition in the discussion section to address this concern.

Second, the authors should revise the Introduction to be more concise and focused. This is not a review paper.

RESPONSE: We shortened the introduction as suggested.

---

## [Decision Letter · Decision Letter 2]

11 Oct 2022

PONE-D-22-12279R2Pain Perception Genes, Asthma, and Oral Health: A Reverse Genetics StudyPLOS ONE

Dear Dr. Vieira,

Thank you for submitting your manuscript to PLOS ONE. After careful consideration, we feel that it has merit but does not fully meet PLOS ONE’s publication criteria as it currently stands. Therefore, we invite you to submit a revised version of the manuscript that addresses the points raised during the review process.

There is one outstanding concern that potentially some citations have been ommitted.

We look forward to receiving your revised manuscript.

Kind regards,

JJ Cray Jr., Ph.D.

Academic Editor

PLOS ONE

Journal Requirements:

Reviewers' comments:

Reviewer's Responses to Questions

**Comments to the Author**

1. If the authors have adequately addressed your comments raised in a previous round of review and you feel that this manuscript is now acceptable for publication, you may indicate that here to bypass the “Comments to the Author” section, enter your conflict of interest statement in the “Confidential to Editor” section, and submit your "Accept" recommendation.

Reviewer #1: All comments have been addressed

Reviewer #2: All comments have been addressed

2. Is the manuscript technically sound, and do the data support the conclusions?

Reviewer #1: Yes

Reviewer #2: Yes

3. Has the statistical analysis been performed appropriately and rigorously? 

Reviewer #1: Yes

Reviewer #2: Yes

4. Have the authors made all data underlying the findings in their manuscript fully available?

Reviewer #1: Yes

Reviewer #2: Yes

5. Is the manuscript presented in an intelligible fashion and written in standard English?

Reviewer #1: Yes

Reviewer #2: Yes

6. Review Comments to the Author

Reviewer #1: The authors have carefully addressed the questions from the reviewers, and significantly improved the quality of the manuscript. However, there are still minor issues that need to be addressed. All the new data included are lacking references. It is necessary to cite the sources from which the information was obtained.

Reviewer #2: (No Response)

7. PLOS authors have the option to publish the peer review history of their article (what does this mean?). If published, this will include your full peer review and any attached files.

Reviewer #1: **Yes: **Flávia Fonseca Carvalho Soares

Reviewer #2: No

---

## [Author Response · Author response to Decision Letter 2]

17 Oct 2022

Reviewer #1: The authors have carefully addressed the questions from the reviewers, and significantly improved the quality of the manuscript. However, there are still minor issues that need to be addressed. All the new data included are lacking references. It is necessary to cite the sources from which the information was obtained.

RESPONSE: We added the references as requested.

---

## [Editor Report · Decision Letter 3]

19 Oct 2022

Pain Perception Genes, Asthma, and Oral Health: A Reverse Genetics Study

PONE-D-22-12279R3

Dear Dr. Vieira,

We’re pleased to inform you that your manuscript has been judged scientifically suitable for publication and will be formally accepted for publication once it meets all outstanding technical requirements.

Kind regards,

JJ Cray Jr., Ph.D.

Academic Editor

PLOS ONE
---

## [Editor Report · Acceptance letter]

9 Nov 2022

PONE-D-22-12279R3 

Pain Perception Genes, Asthma, and Oral Health: A Reverse Genetics Study 

Dear Dr. Vieira:

I'm pleased to inform you that your manuscript has been deemed suitable for publication in PLOS ONE. Congratulations! Your manuscript is now with our production department. 

Kind regards, 

on behalf of

Dr. JJ Cray Jr. 

Academic Editor

PLOS ONE